# Patients with Thyroid Dyshormonogenesis and *DUOX2* Variants: Molecular and Clinical Description and Genotype–Phenotype Correlation

**DOI:** 10.3390/ijms25158473

**Published:** 2024-08-03

**Authors:** Noelia Baz-Redón, María Antolín, María Clemente, Ariadna Campos, Eduard Mogas, Mónica Fernández-Cancio, Elisenda Zafon, Elena García-Arumí, Laura Soler, Núria González-Llorens, Cristina Aguilar-Riera, Núria Camats-Tarruella, Diego Yeste

**Affiliations:** 1Growth and Development Group, Vall d’Hebron Research Institute (VHIR), Hospital Universitari Vall d’Hebron, 08035 Barcelona, Spain; maria.clemente@vallhebron.cat (M.C.); ariadna.campos@vallhebron.cat (A.C.); eduard.mogas@vallhebron.cat (E.M.); monica.fernandez.cancio@vhir.org (M.F.-C.); nucata@yahoo.es (N.C.-T.); diego.yeste@vallhebron.cat (D.Y.); 2Centro de Investigación Biomédica en Red de Enfermedades Raras (CIBERER), Instituto de Salud Carlos III, 28029 Madrid, Spain; elena.garcia@vallhebron.cat; 3Department of Clinical and Molecular Genetics and Rare Disease, Hospital Universitari Vall d’Hebron, 08035 Barcelona, Spain; maria.antolin@vallhebron.cat (M.A.); elisenda94@gmail.com (E.Z.); 4Medicine Genetics Group, Vall d’Hebron Research Institute (VHIR), Hospital Universitari Vall d’Hebron, 08035 Barcelona, Spain; 5Pediatric Endocrinology Section, Hospital Universitari Vall d’Hebron, 08035 Barcelona, Spain; laurasc22@hotmail.com (L.S.); nuria.gonzalezllorens@vallhebron.cat (N.G.-L.); cristina.aguilar@vallhebron.cat (C.A.-R.); 6Pediatrics, Obstetrics and Gynecology and Preventive Medicine Department, Universitat Autònoma de Barcelona, 08193 Bellaterra, Spain

**Keywords:** congenital hypothyroidism, CH, thyroid dyshormonogenesis, dual oxidase 2, *DUOX2*, phenotypic variability

## Abstract

Thyroid dyshormonogenesis (THD) is a heterogeneous group of genetic diseases caused by the total or partial defect in the synthesis or secretion of thyroid hormones. Genetic variants in *DUOX2* can cause partial to total iodination organification defects and clinical heterogeneity, from transient to permanent congenital hypothyroidism. The aim of this study was to undertake a molecular characterization and genotype–phenotype correlation in patients with THD and candidate variants in *DUOX2*. A total of 31 (19.38%) patients from the Catalan Neonatal Screening Program presented with variants in *DUOX2* that could explain their phenotype. Fifteen (48.39%) patients were compound heterozygous, 10 (32.26%) heterozygous, and 4 (12.90%) homozygous. In addition, 8 (26.67%) of these patients presented variants in other genes. A total of 35 variants were described, 10 (28.57%) of these variants have not been previously reported in literature. The most frequent variant in our cohort was c.2895_2898del/p.(Phe966SerfsTer29), classified as pathogenic according to reported functional studies. The final diagnosis of this cohort was permanent THD in 21 patients and transient THD in 10, according to reevaluation and/or need for treatment with levothyroxine. A clear genotype–phenotype correlation could not be identified; therefore, functional studies are necessary to confirm the pathogenicity of the variants.

## 1. Introduction

Congenital hypothyroidism (CH), defined as thyroid hormone deficiency present at birth, is the most frequent neonatal endocrine disorder and one of the most preventable causes of mental disabilities and neurological alterations in children [1,2]. CH is classified as permanent or transient hypothyroidism. Permanent CH is the condition of persistent thyroid hormone deficiency with lifelong necessary treatment. In contrast, transient CH occurs in iodine deficient areas and is characterized by recovery in just a few months or years [2]. CH can be caused by hypothalamus and pituitary disorders (central CH), or due to thyroid gland disfunction (primary CH), which is the most frequent cause of CH, reported in 1:1660 to 1:4000 newborns [3,4,5]. Primary CH is caused by defects in gland development (thyroid dysgenesis, TD) or alterations (partial or total blockage) of the biosynthesis of thyroid hormones (thyroid dyshormonogenesis, THD) [1,2]. THD is a heterogeneous group of genetic diseases, from subclinical hypothyroidism (SH) to goiter [1]. The widely use of high-throughput sequencing techniques and the increasingly precise knowledge of the enzymatic pathways in the synthesis of thyroid hormones have made it possible to identify the transcription factors and proteins involved in these processes. Until now, the THD related factors and their defects include: (1) defects in iodine uptake into thyroid follicular cells due to mutations in *SLC5A5*; (2) iodine organification defects due to mutations in *TPO*, *DUOX2* and *DUOXA2*, encoding the peroxidase enzyme system; (3) defects in iodine transport (Pendred syndrome) due to mutations in the Pendrin gene (*SLC26A4*); (4) defects in thyroglobulin synthesis (*TG*), storage, or secretion; and (5) failure of iodine recycling due to mutations in *IYD* [6]. 

Hydrogen peroxide (H_2_O_2_), generated by oxidases, is essential for the iodination of tyrosil residues of thyroglobulin (Tg), and subsequent biosynthesis of thyroid hormones in the follicular lumen of the thyroid gland by thyroperoxidase (TPO) [2]. Two homologous NADPH oxidases, the dual oxidase 1 and 2 (DUOX1 and DUOX2, respectively) and their maturation factors (DUOXA1 and DUOXA2), have been described as the peroxidase system in the thyroid gland [7]. The dimerization of DUOX-DUOXA is necessary for the correct protein folding and the translocation from endoplasmic reticulum to the plasma membrane [8]. In human thyroid, the *DUOX2* transcript is two to five times more expressed than *DUOX1* [9].

The *DUOX2* gene (NG_009447) is located in the 15q21.1 locus with 21.5 kb and a total of 34 exons (NM_014080.5) [9]. The protein coding sequence (CDS) is composed of 33 coding exons (from exon 2 to 34) and translates in the DUOX2 protein (NP_054799) with a total of 1548 amino acids [10]. DUOX2 is a glycoprotein with seven transmembrane domains [10]. Between the first and second transmembrane domains, three EF-hand motifs for calcium binding and, likely related with enzyme activity regulation, are localized. The extracellular N-terminal segment contains the peroxidase-like domain, and the intracellular C-terminal contains the oxidase domains: the FAD- and NADPH-binding sites [10,11] (Figure 1). As mentioned before, DUOX2 is necessary for H_2_O_2_ generation and thyroid hormones biosynthesis. Therefore, likely pathogenic variants in *DUOX2* (MIM:606759) have been described to cause partial to total iodination defects [1]. Patients with *DUOX2* defects show significantly higher serum thyrotropin (TSH) and thyroglobulin (Tg), and lower free T4 (FT4) [12]. The scintigraphy of these patients presents a high iodine uptake with a 10–90% partial iodine organification defect (PIOD) in the perchlorate discharge test [1]. 

Both, an autosomal recessive and an autosomal dominant inheritance, have been described in THD patients with *DUOX2* candidate variants [1]. Firstly, monoallelic *DUOX2* variants were related with transient and moderate CH, and it has been postulated that biallelic mutations might be associated with permanent (severe or mild) CH forms [13]. However, subsequent studies showed no correlation between patients’ genotype and their phenotype in terms of biochemical hormone tests values and CH duration (i.e., transient or permanent) [11,14,15]. Indeed, monoallelic and biallelic *DUOX2* mutations have been related with permanent or transient CH [11,14]. This variability in *DUOX2* patients’ phenotype was also observed between family members [14,15]. Vigone et al. [15] described the first familial case of patients with CH and *DUOX2* defects: two siblings with the same genotype and relevant phenotypic differences that were attributed to environmental factors (high iodine load in one sibling).

Interestingly, the disease was not detected during the neonatal screening in the sibling with less severe CH. The parents, carrying monoallelic *DUOX2* variants, presented with mild PIOD [15]. Different mechanisms have been postulated to explain this high phenotype variability in patients with CH and *DUOX2* variants [10].

The use of high-throughput sequencing techniques has allowed us to identify the specific genetic cause of THD in patients from the Catalan CH Neonatal Screening Program. In this study, a molecular and clinical characterization and genotype–phenotype correlation in patients with THD and candidate variants in the *DUOX2* were described to understand and predict the disease’s symptoms, progression, and prognosis.

## 2. Results

### 2.1. Genetic Results

A total of 38 patients (23.75% of the THD cohort), including 37 probands, in our Catalan CH Neonatal Screening Program cohort presented variants in the *DUOX2* gene. However, 7 patients were excluded from the study: 5 patients (THD-2, THD-19, THD-20, THD-21, and THD-31) with likely benign variants in the *DUOX2* gene that could not explain their phenotype; patient THD-4 with likely pathogenic variants in *TG* that were considered the cause of their CH; and patient THD-15 because of the lack of clinical data due to loss to follow-up. The clinical and biochemical characteristics and the genetic variants of the excluded patients are described in Appendix A.

Thirty-one patients (19.38%) presented candidate variants in the *DUOX2* gene that could explain their phenotype. Their clinical manifestations and genetic results are described in Table 1. Sixteen of the patients were female, the mean gestational age was 38.51 ± 9.97 weeks, and most of the patients were pediatric with a mean age at study of 7.19 years old (range 1–24 years). The anthropometric neonatal data is described in Appendix A. A total of 15 (48.39%) patients were compound heterozygous, 10 (32.26%) heterozygous, and 4 (12.90%) homozygous. The remaining patients (THD-1 and THD-37) carried two variants, but familial segregation studies would be required to confirm the inheritance. In addition, 8 (26.67%) of these patients presented variants in other genes: 4 patients presented additional variants in *TG* (THD-1, THD-9, THD-10, THD-12), one in *TPO* (THD-8), one in *DUOX1* (THD-17), one in *DUOXA2* (THD-33), and one in *ANO1* (THD-35) (Table 1).

A total of 35 variants were described: 24 missense, 4 nonsense, 3 frameshift, 3 intronic variants with possible splicing defects, and one small deletion (Table 2). Twenty-five of these variants have been previously described in literature associated with CH [13,15,16,17,18,19,20,21,22,23,24,25,26,27,28,29,30,31,32], and 10 (28.57%) were novel. Eight (22.86%) were classified as pathogenic variants, 11 (31.43%) as likely pathogenic, 14 (40%) as variants of uncertain significance (VUS), and 2 (5.71%) as likely benign. To note that, the variants p.(Gln202ThrfsTer99), p.(Gln570Leu), p.(Arg885Gln), p.(Phe966SerfsTer29), p.(Arg1110Gln) were described as pathogenic considering previously published in vitro functional assays showing deleterious activity [18,22,33,34,35]. The most frequent variant in our cohort was c.2895_2898del; p.(Phe966SerfsTer29), a protein truncating variant classified as pathogenic, and observed in 8 of the 31 patients (25.81%) (Table 2). The majority of the exonic variants, including protein truncating variants, are localized in the functional (peroxidase-like and oxidase) domains of the DUOX2 protein. Four variants [p.(Arg861Cys), p.(Asp870Gly), p.(Arg885Gln), and p.(Arg885Leu)] are localized in the EF-hand 2 motif. The remaining exonic variants are localized outside the functional domains. Nevertheless, two frameshift variants [p.(Phe966SerfsTer29) and p.(Phe999LeufsTer26)] and the nonsense variant p.(Gln1301Ter), are predicted to cause loss of C-terminal FAD and NAD binding domains, both necessary for the DUOX2 reaction (Figure 1 and Table 2).

### 2.2. In Silico Studies Results

The canonical splice site variant in patient THD-1 (c.513+1G>C), some protein truncating variants, and several previously described missense variants causing THD in our cohort, were classified as pathogenic and likely pathogenic variants. Furthermore, in silico studies confirmed that the small deletion p.(Phe772del) and missense variants classified as VUS were located in important DUOX2 functional domains, affected a highly conserved amino acid position, and/or decreased the protein stability (Figure 1 and Table 2). Regarding the ten novel variants described in our cohort, only a protein truncating variant was described, the frameshift variant p.(Phe999LeufsTer26) found in homozygosis in patient THD-7 and classified as likely pathogenic (Table 1 and Table 2). The novel missense variants were classified as VUS, but were also located in functional DUOX2 protein domain, affected a conserved amino acid, and/or decrease the protein stability (Table 2). The only exception was the missense variant p.(Tyr185Cys) that affected an amino acid position highly conserved throughout evolution and was classified as likely pathogenic (Table 2). The intronic variants c.514-49G>A and c.4396-14C>T, were classified as likely benign and VUS, respectively, and no splicing defect was predicted by consulted splicing predictors (Table 2).

### 2.3. Patients’ Evaluation, Biochemical Analyses, and Classification

The biochemical thyroid hormone results during the first medical visit (diagnosis confirmation) were a TSH mean value of 158.76 ± 115 mIU/L, mean FT4 levels of 0.86 ± 0.22 ng/dL, and mean serum Tg levels of 1470.36 ± 1044.33 ng/mL.

Twenty-three patients have been reevaluated at a mean age of 3.23 years old (range 1–5) and, considering the TSH and FT4 levels, 8 patients were diagnosed with hyperthyrotropinemia, 9 had transient hypothyroidism, and 6 mild permanent hypothyroidism (Table 1). Based on the need of high levothyroxine doses, patients THD-10, THD-14, THD-18, THD-28, THD-30, THD-34, and THD-38 were reclassified from hyperthyrotropinemia to permanent. The non-revaluated THD-1, THD-11, THD-16, THD-22, THD-29, THD-33, and THD-37 patients needed treatment and were considered permanent. Furthermore, patient THD-24 did not need treatment, so it was not reevaluated, and his THD was considered transient (Table 1). High variability was observed regarding the scintigraphy and ultrasonography scan analyses with 13 hypercaptant patients, 13 normocaptant patients, 2 hypocaptant patients, and one non-captant patient. Ultrasonography scan showed that patient THD-23 presented with hypoplastic gland, and patients THD-5 and THD-8 with goiter. Perchlorate discharge test was available only in 12 patients: 8 patients with negative result (<10%) and 4 patients with a PIOD (10–90%) (Table 1).

## 3. Discussion

A total of 31 (19.38%) of the 160 pediatric patients with THD from the Catalan CH Neonatal Screening Program presented variants in the *DUOX2* gene that could explain their phenotype (Figure 1 and Table 1). The clinical and molecular description and genotype–phenotype correlation were carried out in the present study.

According to previous cohort studies, the prevalence of *DUOX2* variants among CH patients is variable, but generally high. In China, *DUOX2* was considered the most frequently mutated gene, with a prevalence of 60% [30]. The frequency in Japan (43%) was somewhat lower [34]. In Europe, the frequency also varies between different cohorts, for example, 50% in the United Kingdom [36] and 23.33% in Italy [35]. The prevalence in our population (19.38%) was lower than the previously described ones, but more similar to Italy, which could be explained by a different CH genetic background depending on the population. Nevertheless, this prevalence should keep being studied when increasing the population size in order to obtain a more accurate value.

A total of 35 variants were described: 24 missense, 4 nonsense, 3 frameshift, 3 intronic variants with a possible splicing defect, and one small deletion (Table 2). The most frequent *DUOX2* variant in our cohort was c.2895_2898del; p.(Phe966SerfsTer29), observed in 8 (THD-5, THD-6, THD-13, THD-16, THD-18, THD-25, THD-26, THD-38) of the 31 patients (25.81%) (Table 2). This frameshift variant was classified as pathogenic, according to ACMG criteria, and was firstly described by Moreno et al. [13] in a heterozygous patient with transient CH and partial iodine organification defect [13]. In a CH cohort study by Muzza et al. [12], this variant was also the most frequent one, explaining the 14% of *DUOX2* patients, who presented a high clinical variability [12]. Furthermore, functional studies of this variant were carried out in homozygosis by De Marco et al. [35], demonstrating that it causes a decreased expression of the DUOX2 protein in the cellular membrane and the complete inhibition of H_2_O_2_ production [35]. In our cohort, 3 patients (THD-13, THD-18, and THD-25) were homozygous for this variant, and the other 5 patients were compound heterozygous (THD-5, THD-6, THD-16, THD-26, and THD-38) carrying other candidate *DUOX2* variants. The phenotype of these patients, even the 3 homozygous, was variable and does not seem to be directly related with their genotype, as previously described in the literature [12].

The majority of the variants described in our cohort were missense, a total of 24 out of 35 variants. According to the *DUOX2* metric parameters, the gene is tolerant for missense mutations [z-score = (−1.17)]. Nevertheless, the variants p.(Gln570Leu), p.(Arg885Gln), and p.(Arg1110Gln) were previously functionally described to reduce the H_2_O_2_ production and should be considered pathogenic [22,33,34]. Among these variants, p.(Arg885Gln) and p.(Arg1110Gln) were firstly described in patients with an autosomal recessive inheritance, and diagnosed with transient and permanent CH, respectively [21,26]. The CH reevaluation results of these reported patients was the same as those patients of our cohort carrying the same variants (THD-12 and THD-11). Most of these missense variants, including novel missense variants, were classified as VUS, but were considered in this study due to the patients’ CH phenotype and the in silico results (Table 1 and Table 2). To note that, 7 of these variants were previously described in other CH cohorts (Table 2). The variant p.(Ala1131Ser) was classified as VUS, but previously described in a compound heterozygous patient with permanent CH, like the one in our cohort (THD-28) [27]. The phenotype of the patients carrying previously described variants presented with different variant inheritance and/or CH phenotype than our patients [15,16,23]. Patient THD-22 carried three *DUOX2* variants [p.(Arg683Leu)/p.(Phe772del)/p.(Leu1343Phe)], two of them missense variants [p.(Arg683Leu) and p.(Leu1343Phe)] classified as VUS and likely benign, respectively (Table 1 and Table 2). Both variants were previously described by Fu et al. [23] in a compound heterozygous patient with triallelic *DUOX2* variants and permanent CH [23]. The phenotype of the reported patient is similar to ours, diagnosed with permanent CH due to the need of high levothyroxine doses. Therefore, it would be of high interest to carry out in vitro functional studies to confirm the pathogenicity of VUS missense variants.

Regarding the intronic variants described in our cohort, c.513+1G>C (carried by THD-1) affected a canonical splice site and was considered pathogenic. The variant c.4396-14C>T (THD-35) was predicted not to affect splicing, according to in silico predictors, and classified as VUS. This variant was carried in patient THD-35 in compound heterozygosity with a pathogenic variant c.1275T>G;p.(Thr425Ter) and an additional *ANO1* variant, which could be coherent with his transient THD phenotype (Table 1). Patient THD-10 was heterozygous for c.514-49G>A in *DUOX2*, classified as likely benign, but also carried an intronic *TG* variant (c.5975+5G>C) that was moderately predicted to cause a donor splice site loss (SpliceAI Δ score = 0.76) and could explain the patient’s mild permanent hypothyroidism diagnosis (Table 1). In vitro RNA functional studies would be necessary to confirm the splicing effect of these variants and the digenic cause in patient THD-10.

Eight (25.81%) patients of our *DUOX2* cohort presented variants in other THD causing genes, suggesting a possible digenic inheritance for the disease: 4 patients presented variants in *TG* (THD-1, THD-9, THD-10, THD-12), one in *TPO* (THD-8), one in *DUOX1* (THD-17), one in *DUOXA2* (THD-33), and one in *ANO1* (THD-35) (Table 1). Patient THD-4 was excluded as we considered the two additional *TG* variants the cause of her phenotype, according to clinical manifestation with low Tg levels (0.3 ng/mL), and the variants’ classification as pathogenic (Appendix A). Regarding patients with *DUOX2* and *TG* candidate variants (Table 1), patient THD-12 was compound heterozygous for two missense *DUOX2* variants [p.(Arg885Leu)/p.(Arg885Gln)] classified as likely pathogenic and pathogenic, respectively. This patient also carried a *TG* missense variant and presented with high levels of Tg (550 ng/mL). Therefore, the *DUOX2* variants might be the cause of the patient’s phenotype, as the *TG* variant does not seem to worsen the transient THD. Patient THD-1 presented two likely pathogenic variants in *DUOX2* and one missense benign variant in *TG* [c.3749G>T;p.(Arg1250Leu)]; and, although the familial segregation study would be necessary to describe the zygosity, it seems that *DUOX2* could be the CH causing gene. Patients THD-9 and THD-10 presented monoallelic variants in both genes, *DUOX2* and *TG*, and the CH phenotypes have been related to a possible digenic cause. Patient THD-8 presented monoallelic variants in *DUOX2* and *TPO*. Even though no previous cases with this digenic inheritance were reported, to our knowledge, it would be expected that these cases would have a more severe phenotype, as *TPO* mutations have been associated with complete iodine organification defects and severe CH [1]. Patient THD-8 was heterozygous for a VUS missense *DUOX2* variant p.(Asp115Tyr) and *TPO* variant c.1339-58G>A, which was predicted to not affect splicing. The patient presented with mild permanent CH, so functional studies of the variants would be of high interest (Table 1). Patient THD-17 presented two *DUOX2* missense variants [p.(Gln570Leu)/p.(Arg861Cys)] and a splicing variant in *DUOX1* (c.2548+1G>T). It has been speculated that the H_2_O_2_ production by the isoenzyme DUOX1 might variably compensate for the DUOX2 deficiency. Furthermore, Aycan et al. [37] first described a familial case of two siblings with *DUOX2* and *DUOX1* mutations and a more severe phenotype with total iodine organification defect [37]. Considering these previously reported data, the transient phenotype of our patient THD-17 would not concur, and functional studies of the VUS p.(Arg861Cys) variant would be necessary to confirm its pathogenicity. Patient THD-35 carried biallelic variants in *DUOX2* (c.4396-14C>T/c.1275T>G) and a missense variant in *ANO1* [c.1052T>G;p.(Ile351Thr)], classified as VUS (Table 1). The *ANO1* gene encodes the Anoctamin 1 (ANO1) protein, calcium-activated chloride channel that may contribute to the rapid iodine efflux for the synthesis of thyroid hormones [38]. To our knowledge, no patients with *DUOX2* and *ANO1* variants were described in the literature, but considering the role of ANO1 in thyrocytes, variants in its gene may contribute in the CH patients’ phenotype.

The *DUOX2* patients in our cohort presented with elevated serum TSH concentrations 158.76 ± 115 mIU/L (reference range 0.87–6.15 mIU/L) as a characteristic of their CH, low FT4 levels 0.86 ± 0.22 ng/dL (reference values 0.94–1.9/0.94–1.44 ng/dL) (Table 1), as described in previously reported *DUOX2* patients [12]. Moreover, all the patients in our cohort presented with high mean Tg levels of 1470.36 ± 1044.33 ng/mL (reference values 3.5–77 ng/mL) (Table 1), as previously described in other cohort studies [13] and consistent with iodine organification defects [39].

Most of the patients with *DUOX2* variants described in this study do not present clear genotype–phenotype correlation, which is concordant with previous publications [11,14,15]. Different mechanisms have been postulated to explain this phenotype variability: (1) partially compensatory DUOX1 activity in *DUOX2* affected patients, even though *DUOX1* transcript is less expressed in thyroid than *DUOX2* [10,14]; (2) H_2_O_2_ production by NOX4, although its localization has been described to be mainly intracellular [10,14]; (3) variable thyroid hormone requirement with age, which progressively declines after high levels at birth and transiently increases in puberty or pregnancy [10,14]; and (4) dietary iodine intake [10,14].

In conclusion, no genotype–phenotype correlation could be determined in patients with THD and *DUOX2* variants from the Catalan CH Neonatal Screening Program. That is why, reevaluation of these patients and their close monitoring would be necessary even after reaching an euthyroid state.

## 4. Materials and Methods

### 4.1. Patients and Clinical Evaluation

A total of 160 patients with THD were included in the study from the CH Neonatal Screening Program in Catalonia and treated at the Pediatric Endocrinology Unit of the Vall d’Hebron Universitary Hospital (HUVH) between 2011 and 2022. The CH prevalence in Catalonia is 1:2305 [40]. This study is an ambispective, retrospective and longitudinal study. All patients, or their responsible guardians, provided informed consent for the use of their samples in research studies. Their confidentiality was maintained by assigning a sample code. Our project was approved by the Clinical Research Ethics Committee (CEIC) of the HUVH (PR (AMI) 390/2016).

THD was defined as plasma TSH levels at neonatal evaluation or re-testing greater than 10 mIU/L (AutoDELFIA^®^, Perkin-Elmer, Waltham, MA, USA). Clinical and biochemical data were collected in the first medical visit after receiving the CH positive results in the neonatal screening program to confirm the diagnosis. The thyroid scintigraphy was also carried out in the first medical visit, and in patients with an absence of thyroid uptake, an ultrasonography scan was performed to confirm the absence or presence of thyroid tissue. Additional information on the possible existence of thyroid disease in members of the family or associated malformations/diseases was collected in all cases. Patients with serum TSH levels below 10 mIU/L, with thyroid agenesis or ectopia diagnosed by thyroid scintigraphy and/or ultrasound, or with recognized syndromes were excluded.

Serum TSH levels at confirmation were determined by electrochemiluminescence immunoassay (TSH3-Ultra, ADVIA Centaur^®^, Siemens Healthineers, Erlangen, Germany), free T4 (fT4) levels were determined by electrochemiluminescence immunoassay (FT4, ADVIA Centaur^®^), and TG levels were determined by electrochemiluminescence immunoassay (Elecsys^®^ Tg II-Cobas^®^, Roche, Basel, Switzerland).

Permanent or transient CH was determined using results of thyroid function tests after temporary withdrawal of LT4 therapy at approximately 3 years of age. One month after discontinuing levothyroxine (LT4) treatment, TSH and fT4 levels were measured in a venous blood sample. Individuals who showed continuous dependency on LT4 were diagnosed with permanent CH. After reevaluation, patients with TSH ≥ 10 mIU/L and fT4 < 0.8 ng/dL were classified as severe permanent THD and patients with TSH ≥ 10 mIU/L and fT4 > 0.8 ng/dL as mild permanent THD. Subsequently, these children were then repeatedly evaluated at regular intervals of six months to monitor thyroid function. Those who did not need continuous LT4 therapy were diagnosed with transient CH (TSH < 5 mIU/L and fT4 > 0.8 ng/dL). Reevaluation was not carried out when LT4 requirements (>3.4 mcg/day) and genetic results were suggestive of *DUOX2* deficiency.

Perchlorate discharge tests were carried out, when possible, during reevaluation by the Nuclear Medicine Department of the HUVH. To perform the test, the radiotracer I^123^ was first administered to the patient and anterior cervical thyroid images were obtained after 2 h. Afterwards, 500 mg of sodium perchlorate were administered orally and new thyroid images were obtained 30 and 60 min after its administration, and the percentage of thyroid clearance was quantified according to the baseline image. The results of this test were classified as: negative test when ≤10% clearance, PIOD when 10–90% clearance, and total iodine organification defect when ≥90% clearance.

In this study, patients with candidate variants in the *DUOX2* gene that could explained their phenotype were included and described. Patients with clinical data not available and likely benign variants that could not explain the THD, were excluded.

### 4.2. Genetics

Peripheral blood DNA was extracted by automated magnetic extraction (Chemagic 360, Perkin-Elmer). DNA concentration was measured by Qubit 2.0 fluorometer (Qubit dsDNA BR Assay; Thermo Fisher, Waltham, MA, USA). To perform the genetic study, a CH high-throughput custom panel was designed including the main genes involved in the regulation of thyroid hormonogenesis: *DUOX2*, *DUOXA2*, *IYD*, *PAX8*, *TG*, *TPO*, *TSHR*, *SLC26A4*, and *SLC5A5*.

This panel consists of 393 amplicons (with an average size of 194 bp, coverage ± 20 nucleotides of the exon-intronic junction zone) and allows sequencing of exons and adjacent intronic gene regions of interest, with a total coverage of 96.4% of the target sequence. The CH panel was performed on the patients’ DNA samples following the manufacturer’s instructions and, finally, sequenced on a MiSeq platform (Illumina, San Diego, CA, USA). Bioinformatics analysis of the obtained data was performed following the previously reported pipeline [41] and the pathogenicity of the variants was assessed following the American College of Medical Genetics and Genomics (ACMG) criteria (https://www.acmg.net/, accessed on 31 July 2023) [42]. When a genotype diagnosis was not obtained, an additional panel (Cell3 Target Custom Panel tier 2, NONACUS), which included *DUOX1* and *ANO1* genes, was carried out. Variant nomenclature was done following the Human Genome Variation Society guidelines (https://www.hgvs.org/, accessed on 31 July 2023) [43].

Candidate variants, as well as target regions with low coverage (<20×), were analyzed by Sanger sequencing. Familial segregation studies were carried out when possible.

### 4.3. In Silico Studies

To analyze the possible pathogenicity of candidate variants, in silico studies were performed. The *DUOX2* gene metric parameters were consulted in the Genome Aggregation Database (https://gnomad.broadinstitute.org/, accessed on 14 July 2023). The localization of the affected amino acid in the different protein domains was determined using the DUOX2 protein isoform NP_054799.4 and the Q9NRD8 human DUOX2 model in UniProt protein database (https://www.uniprot.org/, accessed on 14 July 2023) [44]. The amino acid conservation through evolution was studied by Clustal Omega (https://www.ebi.ac.uk/Tools/msa/clustalo/, accessed on 14 July 2023) [45,46], and the prediction of protein stability changes caused by the candidate variants using the I-Mutant2.0 website tool (https://folding.biofold.org/i-mutant/i-mutant2.0.html, accessed on 14 July 2023) [47].

For intronic variants and exonic missense variants with localization at 5′ and 3′ ends, the splicing effects were predicted using the website tool SpliceAI (https://spliceailookup.broadinstitute.org/, accessed on 31 July 2023).

## Figures and Tables

**Figure 1 ijms-25-08473-f001:**
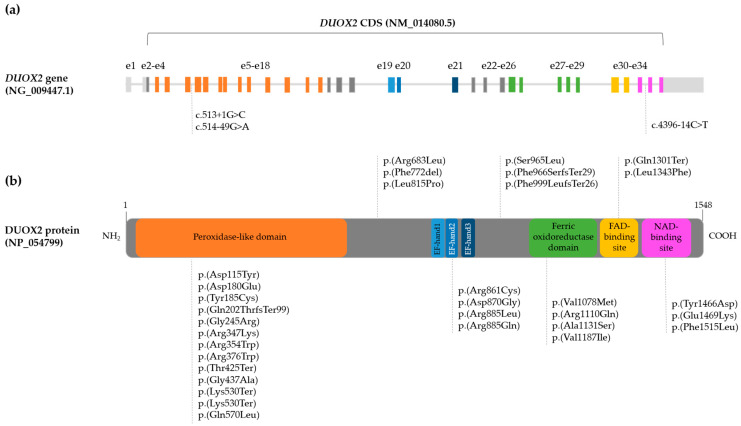
Diagram of *DUOX2* gene and protein. (**a**) *DUOX2* gene (NG_009447.1) with boxes indicating the 34 exons (e1–e34). The coding sequence (CDS) from e2 to e34 (NM_014080.5) codifies for the DUOX2 protein. The dashed lines indicate the intronic *DUOX2* variants described in our cohort that explained the phenotype of patients. (**b**) DUOX2 protein (NP_054799) with the corresponding functional domains (colored boxes). The dashed lines indicate the localization of the exonic variants detected in our cohort.

**Table 1 ijms-25-08473-t001:** Clinical manifestations and genetic results of patients with confirmed thyroid dyshormonogenesis and candidate variants in the *DUOX2* gene.

Patient	TSH Neonatal Screening (mIU/L)	TSH Conf (mIU/L)	FT4 Conf (ng/dL)	Tg Conf (ng/mL)	Scintigraphy/Ultrasonography	Perchlorate Discharge Test	Final CH Diagnosis	Nucleotide Change (NM_014080.4)	AA Change (NP_054799.4)	Pathogenicity (ACMG)	Zygo-Sity	Familial Segregation	Variants in Other Genes
TDH-1	65.4	26.8	0.3	NA	hypocaptant/NA	NA	Permanent ^2^	c.513+1G>C/ c.1060C>T	-/p.(Arg354Trp)	P/LP	NA	NA	*TG* [c.3749G>T; p.(Arg1250Leu)]
TDH-3	116	375	0.3	1500	normocaptant/NA	0% (negative)	Mild permanent ^1^	c.554A>G/ c.4405G>A	p.(Tyr185Cys)/ p.(Glu1469Lys)	LP/LP	CompHet	Mo/Fa	
TDH-5	108	295	0.4	NA	normocaptant/goiter	68% (PIOD)	Mild permanent ^1^	c.2895_2898del/c.3901C>T	p.(Phe966SerfsTer29)/p.(Gln1301Ter)	P/P	CompHet	Mo/Fa	
TDH-6	51.5	294	0.3	>1500	normocaptant/NA	<10% (negative)	Transient ^1^	c.1060C>T/ c.2895_2898del	p.(Arg354Trp)/ p.(Phe966SerfsTer29)	LP/P	CompHet	Fa/Mo	
TDH-7	56	257	0.65	1357	hypercaptant/NA	40% (PIOD)	Mild permanent ^1^	c.2997delT/ c.2997delT	p.(Phe999LeufsTer26)/ p.(Phe999LeufsTer26)	LP/LP	Hom	Fa/Mo	
TDH-8	24.3	17.1	1.2	113	normocaptant/goiter	74% (PIOD)	Hyperthyro-tropinemia ^1^	c.343G>T	p.(Asp115Tyr)	VUS	Het	Fa	*TPO* [c.1339-58G>A]
TDH-9	12	56.8	0.9	NA	normocaptant/normal	0% (negative)	Mild permanent ^1^	c.540C>A	p.(Asp180Glu)	VUS	Het	Fa	*TG* [c.5402-8C>T]
TDH-10	60	231	0.55	203.1	hypercaptant/NA	NA	Permanent ^2^	c.514-49G>A		LB	Het	NA	*TG* [c.5975+5G>C]
TDH-11	28	35.4	1.5	488	hypercaptant/normal	NA	Permanent ^2^	c.1588A>T/ c.3329G>A	p.(Lys530Ter)/ p.(Arg1110Gln)	P/P	CompHet	Mo+sib/- (Fa NA)	
TDH-12	75	23.7	1.2	550	hypercaptant/NA	0% (negative)	Transient ^1^	c.2654G>T/ c.2654G>A	p.(Arg885Leu)/ p.(Arg885Gln)	LP/P	CompHet	Mo+sib/Fa	*TG* [c.4270G>T; p.(Phe1424Val)]
TDH-13	39.2	17.1	1.3	1120	hypercaptant/NA	0% (negative)	Transient ^1^	c.2895_2898del/c.2895_2898del	p.(Phe966SerfsTer29)/ p.(Phe966SerfsTer29)	P/P	Hom	Fa/Mo	
TDH-14	22	50.3	1.45	536	normocaptant/NA	0% (negative)	Permanent ^2^	c.602dupG	p.(Gln202ThrfsTer99)	P	Het	Fa	
TDH-16	14	50.9	0.7	1370	hypercaptant/NA	NA	Permanent ^2^	c.2444T>C/ c.2895_2898del	p.(Leu815Pro)/ p.(Phe966SerfsTer29)	LP/P	CompHet	Fa/Mo	
TDH-17	76	49.8	0.7	1724	hypercaptant/NA	0% (negative)	Transient ^1^	c.1709A>T/c.2581C>T	p.(Gln570Leu)/ p.(Arg861Cys)	P/VUS	CompHet	Mo/Fa	*DUOX1* [c.2548+1G>T]
TDH-18	47	289	0.53	4588	hypercaptant/NA	0% (negative)	Permanent ^2^	c.2895_2898del/c.2895_2898del	p.(Phe966SerfsTer29)/ p.(Phe966SerfsTer29)	P/P	Hom	Fa/Mo	
TDH-22	44.6	56.6	1.2	414.1	hypercaptant/NA	NA	Permanent ^2^	c.2048G>T/ c.2314_2316del/c.4027C>T	p.(Arg683Leu)/ p.(Phe772del)/ p.(Leu1343Phe)	VUS/VUS/LB	CompHet	Mo/Fa/Mo	
TDH-23	22.6	23.4	1.6	128.5	non-captant/ hypoplasia	NA	Transient ^1^	c.3559G>A	p.(Val1187Ile)	VUS	Het	NA	
TDH-24	41	24.7	1.00	419.2	normocaptant/NA	NA	Transient ^2^	c.1709A>T	p.(Gln570Leu)	P	Het	NA	
TDH-25	75	438.1	0.1	1185	hypercaptant/NA	NA	Mild permanent ^1^	c.2895_2898del/c.2895_2898del	p.(Phe966SerfsTer29)/ p.(Phe966SerfsTer29)	P/P	Hom	Fa/Mo	
TDH-26	16	19.5	1.12	1361	hypercaptant/NA	NA	Transient ^1^	c.2895_2898del/c.4396T>G	p.(Phe966SerfsTer29)/ p.(Tyr1466Asp)	P/VUS	CompHet	Fa/- (Mo NA)	
TDH-27	NA	23	1.29	NA	NA/normal	NA	Transient ^1^	c.3232G>A	p.(Val1078Met)	VUS	Het	NA	
TDH-28	10.3	35	0.62	NA	NA/NA	NA	Permanent ^2^	c.3232G>A/c.3391G>T	p.(Val1078Met)/ p.(Ala1131Thr)	VUS/VUS	CompHet	-/Mo (Fa NA)	
TDH-29	42.7	225.85	0.31	2477	normocaptant/NA	NA	Permanent ^2^	c.733G>A/ c.1040G>A	p.(Gly245Arg)/ p.(Arg347Lys)	VUS/LP	CompHet	Mo/Fa	
TDH-30	11.1	12.8	1.28	137.6	normocaptant/NA	NA	Permanent ^2^	c.4543T>C	p.(Phe1515Leu)	VUS	Het	Fa	
TDH-32	21.3	99.5	0.72	665.6	normocaptant/NA	NA	Mild permanent ^1^	c.2609A>G	p.(Asp870Gly)	VUS	Het	NA	
TDH-33	11.2	68	0.69	728.1	hypercaptant/NA	NA	Permanent ^2^	c.2654G>T	p.(Arg885Leu)	LP	Het	Mo	*DUOXA2* [c.413dup; p.(Tyr138Ter)]
TDH-34	16.0	51	1.12	NA	hypercaptant/NA	NA	Permanent ^2^	c.1310G>C/ c.1588A>T	p.(Gly437Ala)/ p.(Lys530Ter)	LP/P	CompHet	-/Mo (Fa NA)	
TDH-35	10	22	1.11	401.7	hypocaptant/NA	NA	Transient ^1^	c.1275T>G/ c.4396-14C>T	p.(Thr425Ter)/-	LP/VUS	CompHet	Fa/Mo	*ANO1* [c.1052T>G; p.(Ile351Thr)]
TDH-36	53.2	76.86	1.19	2216	normocaptant/NA	NA	Permanent ^2^	c.1126C>T/ c.2581C>T	p.(Arg376Trp)/ p.(Arg861Cys)	VUS/VUS	CompHet	-/Mo (Fa NA)	
TDH-37	10.5	65.1	0.67	NA	normocaptant/NA	NA	Permanent ^2^	c.1708C>T/ c.2654G>T	p.(Gln570Ter)/ p.(Arg885Leu)	LP/LP	NA	NA	
TDH-38	NA	58.8	0.71	NA	normocaptant/NA	17% (PIOD)	Permanent ^2^	c.2894C>T/ c.2895_2898del	p.(Ser965Leu)/ p.(Phe966SerfsTer29)	LP/P	CompHet	-/Mo (Fa NA)	

Conf: diagnostic confirmation values; TSH: thyroid-stimulating hormone; FT4: free thyroxine; Tg: thyroglobulin; ref: reference values; CH: congenital hypothyroidism; AA: amino acid; NA: non available; NR: non reevaluated; PIOD: partial iodine organification defect; Y: yes; MS: missense; SP: splicing; FS: frameshift; NS: nonsense; DEL: deletion; P: pathogenic; LP: likely pathogenic; VUSvariant of uncertain significance; Het: heterozygous; CompHet: compound heterozygous; Hom: homozygous: Mo: carrier mother; Fa: carrier father; sib: carrier sibling. ^1^ Final diagnosis classification according to TSH and FT4 levels at reevaluation (TSH < 5 mIU/L transient hypothyroidism; TSH 5–10 mIU/L hyperthyrotropinemia; TSH > 10 mIU/L, FT4 normal mild permanent hypothyroidism; TSH > 10 mIU/L, FT4 < 0.8 ng/dL severe permanent hypothyroidism). ^2^ Classification of congenital hypothyroidism severity according to the need for levothyroxine treatment.

**Table 2 ijms-25-08473-t002:** *DUOX2* variants and their in silico studies described in our cohort.

Intron/Exon Variant Localization	Nucleotide Change (NM_014080.4)	AA Change (NP_054799.4)	#Pat (Table 1)	Variant Type	Previously Described Variants ^1^ (Last Revision May 2024), Reference	Reported Inheritance and CH Diagnosis	Pathogenicity (Last Revision July 2023)	DUOX2 Domain	Evolutionary Conservation of AA	Protein Stability (DDG, I-Mutant 2.0 Sequence)
Exon 5	c.343G>T	p.(Asp115Tyr)	THD-8	MS	described, [16]	AR, transient	VUS	PR	+	−0.29
IVS 5	c.513+1G>C		THD-1	IN	described, [17]	ND	P			
	c.514-49G>A		THD-10	IN	novel		LB			
Exon 6	c.540C>A	p.(Asp180Glu)	THD-9	MS	novel		VUS	PR	+++	−0.3
	c.554A>G	p.(Tyr185Cys)	THD-3	MS	novel		LP	PR	+++	0.50
	c.602dupG	p.(Gln202ThrfsTer99)	THD-14	FS	described, [18]	AD, transient	P ^2^ [18]	PR	-	
Exon 7	c.733G>A	p.(Gly245Arg)	THD-29	MS	novel		VUS	PR	++	−0.73
Exon 9	c.1040G>A	p.(Arg347Lys)	THD-29	MS	described, [19]	ND	LP	PR	+++	−1.41
Exon 10	c.1060C>T	p.(Arg354Trp)	THD-1, THD-6	MS	described, [29]	AD, transient	LP	PR	-	−0.65
	c.1126C>T	p.(Arg376Trp)	THD-36	MS	described, [15]	AR, mild CH	VUS	PR	+++	−0.75
Exon 12	c.1275T>G	p.(Thr425Ter)	THD-35	NS	described, [20]	AR, transient	LP	PR	++	
	c.1310G>C	p.(Gly437Ala)	THD-34	MS	described, [30]	ND	LP	PR	+++	0.16
Exon 14	c.1588A>T	p.(Lys530Ter)	THD-11, THD-34	NS	described, [21]	AR, transient	P	PR	-	
Exon 15	c.1708C>T	p.(Gln570Ter)	THD-37	NS	described, [31]	ND	LP	PR	+++	
	c.1709A>T	p.(Gln570Leu)	THD-17, THD-24	MS	described, [22]	AR, permanent	P ^2^ [22]	PR	+++	0.48
Exon 17	c.2048G>T	p.(Arg683Leu)	THD-22	MS	described, [23]	AR, permanent	VUS	IC	+++	−0.46
Exon 18	c.2314_2316delTTC	p.(Phe772del)	THD-22	DEL	described, [30]	ND	VUS	IC	++	
Exon 19	c.2444T>C	p.(Leu815Pro)	THD-16	MS	novel		LP	IC	+++	1.41
Exon 20	c.2581C>T	p.(Arg861Cys)	THD-17, THD-36	MS	described, [24]	ND	VUS	EF-hand 2	+	−1.15
	c.2609A>G	p.(Asp870Gly)	THD-32	MS	novel		VUS	EF-hand 2	++	−0.59
	c.2654G>T	p.(Arg885Leu)	THD-12, THD-33, THD-37	MS	described, [25]	ND	LP	EF-hand 2	+++	−0.76
	c.2654G>A	p.(Arg885Gln)	THD-12	MS	described, [21]	AR, transient	P ^2^ [33]	EF-hand 2	+++	−1.81
Exon 22	c.2894C>T	p.(Ser965Leu)	THD-38	MS	described, [25]	ND	LP	IC	+++	
	c.2895_2898delGTTC	p.(Phe966SerfsTer29)	THD-5, THD-6, THD-13, THD-16, THD-18, THD-25, THD-26, THD-38	FS	described, [13]	AD, transient	P ^2^ [35]	IC	++	
Exon 23	c.2997delT	p.(Phe999LeufsTer26)	THD-7	FS	novel		LP	IC	-	
Exon 25	c.3232G>A	p.(Val1078Met)	THD-27, THD-28	MS	described, [19]	ND	VUS	TM3-TM4	++	−2.52
	c.3329G>A	p.(Arg1110Gln)	THD-11	MS	described, [26]	AR, permanent	P ^2^ [34]	TM3-TM4	+++	−1.57
	c.3391G>T	p.(Ala1131Ser)	THD-28	MS	described, [27]	AR, permanent	VUS	TM4	+++	−0.05
Exon 27	c.3559G>A	p.(Val1187Ile)	THD-23	MS	described, [19]	ND	VUS	TM5	++	−1.36
Exon 30	c.3901C>T	p.(Gln1301Ter)	THD-5	NS	described, [28]	ND	P	FAD	+++	
	c.4027C>T	p.(Leu1343Phe)	THD-22	MS	described, [23]	AR, permanent	LB	FAD	+++	−0.67
IVS 32	c.4396-14C>T		THD-35	IN	novel		VUS			
Exon 33	c.4396T>G	p.(Tyr1466Asp)	THD-26	MS	novel		VUS	NAD	+++	−1.25
	c.4405G>A	p.(Glu1469Lys)	THD-3	MS	described, [32]	AR	LP	NAD	+++	−1.00
Exon 34	c.4543T>C	p.(Phe1515Leu)	THD-30	MS	novel		VUS	NAD	+++	−2.03

AA: amino acid; #Pat: number of patient with the variant; CH: congenital hypothyroidism; IN: intronic variants; MS: missense; FS: frameshift; NS: nonsense; DEL: deleción; AR: autosomal recessive inheritance; AD: autosomal dominant inheritance; ND: no data reported; P: pathogenic; LP: likely pathogenic; VUS: variant of uncertain significance; PR: peroxidase-like domain; IC: intracellular/cytoplasmic; TM: transmembrane; FAD: FAD-binding domain; NADPH: NADPH-binding domain; +++: amino acid position conserved in all studied species; ++: one specie with different amino acid in the same position; +: two species with different amino acid in the same position; -: three or more species with different amino acid in the same position. Highlighted variants with pathogenic functional studies described. DDG > 0: increased protein stability; DDG < 0: decreased protein stability. ^1^ Previously described variants according to Human Gene Mutation Database (HGMD). ^2^ Variants than can be considered pathogenic due to deleterious functional assays reported, references cited in the table.

## Data Availability

The original contributions presented in the study are included in the article/Appendix A; further enquiries can be directed to the corresponding author.

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
