# Peer review of "Patients with Thyroid Dyshormonogenesis and DUOX2 Variants: Molecular and Clinical Description and Genotype–Phenotype Correlation"

_ijms, 2024, doi:10.3390/ijms25158473_

Round 1

Reviewer 1 Report

Comments and Suggestions for Authors

In the manuscript, entitled »Patients with thyroid dyshormonogenesis and DUOX2 variants: molecular and clinical description and genotype-phenotype correlation« submitted to International Journal of Molecular Sciences for a potential publication, the authors present their research article investigating genetic variants in thyroid dyshormonogenesis and genotype phenotype correlations.  I am of opinion, that in the present form, the manuscript is not good enough to be published and needs some revision. My comments to potentially improve it are enclosed.

The comments:

1.      The time of the recruitment of patients in the presented study should be defined clearly (was it from 2001 till 2023?) with the regional prevalence calculated.

2.      Clinical evaluation, not uniformly performed in all patients, should be explained.

3.      The presentation of genetic variants in other three quarters of patients in addition to DUOX2  with their phenotypes in the Catalan cohort would be of value.

4.      The clinical significance in other variants in addition to the DUOX2 in eight out of 31 patients should be presented more cleary and in more detail.

5.      Discussion should be focused in the ten novel variants not previously described in the literature.

6.      Quite low frequency od DUOX2 variants in presented cohort of patients compared to the other populations should be discussed.

7.      Treatment as well as follow-up of the included patients should be presented in more detail.

8.      What would be the clinical application of the study results in the light of the neonatal screening?

9.      Future research needed in the area should be delineated.

Comments on the Quality of English Language

Minor editing of English language is required.

Author Response

Comments:

  1.  The time of the recruitment of patients in the presented study should be defined clearly (was it from 2001 till 2023?) with the regional prevalence calculated.

Authors: We apologize for the mistake, the time of recruitment of patients in the presented study was 2011-2022, this error was corrected in line 326. The regional prevalence was calculated in a publication of the Neonatal Screening Program in Catalonia and it is 1:2,305. We added this information and reference in the manuscript (lines 326-327, reference Marín Soria et al. 2020).

  1. Clinical evaluation, not uniformly performed in all patients, should be explained.

Authors: We beg to differ with the fact that the clinical evaluation was not performed uniformly in all patients, except for the perchlorate discharge test. We performed the same clinical evaluation in all patients, as presented in Table 1. The perchlorate test was performed only in some patients considering their reevaluation and molecular diagnosis.   

  1. The presentation of genetic variants in other three quarters of patients in addition to DUOX2 with their phenotypes in the Catalan cohort would be of value.

Authors: Thank you for the suggestion. We are preparing a manuscript with the genetic results and phenotype descriptions in the Catalan cohort patients. These results will be published when the reevaluation and follow-up of some patients will be completed. However, we decided to publish the DUOX2 results in a communication in order to let the scientific community know the importance of following-up these patients independently of their genotype.

  1. The clinical significance in other variants in addition to the DUOX2 in eight out of 31 patients should be presented more cleary and in more detail.

Authors: Eight patients of our DUOX2 cohort presented with candidate variants in other thyroid dyshormonogenesis causal genes. The variants are presented in Table 1 and discussed in lines 263-301. In these lines the classification of the variants was described and possible digenic inheritances were discussed. 

  1. Discussion should be focused in the ten novel variants not previously described in the literature.

Authors: Thank you for the suggestion, we added a paragraph in the results with the information included in Table 2 (lines 167-176) and the discussion related with these variants was described in lines 236-239, and 253-261.

  1. Quite low frequency od DUOX2 variants in presented cohort of patients compared to the other populations should be discussed.

Authors: The explanation for the low frequency of DUOX2 variants in our cohort was improved (lines 206-210).

  1. Treatment as well as follow-up of the included patients should be presented in more detail.

Authors: Thank you for the suggestion. We have described in detail the treatment and follow-up of the included patients in Materials and Methods (lines 328-331, 333, and 337-356).

  1. What would be the clinical application of the study results in the light of the neonatal screening?

Authors: The molecular diagnosis of the thyroid dyshormonogenesis in patients of the Catalan Congenital Hypothyroidism Neonatal Screening Program allows us to know the specific genetic cause of the disease and correlate it with the patients’ phenotype. In the case of DUOX2, this genotype-phenotype correlation does not exist, which makes the reevaluation and monitoring of patients essential. Our conclusion is presented in lines 318-321.

  1. Future research needed in the area should be delineated.

Authors: Thank you for the comment. We described through the manuscript the importance of future research regarding functional studies of variants of interest (abstract lines 35-36, discussion lines 250-251, 261-262, and 285-287)

Comments on the Quality of English Language: Minor editing of English language is required.

Authors: We thank the reviewer for the comment, we revised the manuscript and corrected some typos and grammatical mistakes.

Reviewer 2 Report

Comments and Suggestions for Authors

Dear Authors,

Thank you for the opportunity to review your interesting manuscript. I find the topic compelling and the paper clear and well-structured. I would like to suggest a few minor modifications, as I noticed some references are missing in the text.

For example, lines 51-52; 62-64; 71-72; 83-84. Line 140: “A total of 35 variants were described: 24 missense, four nonsense, three frameshift, three intronic variants with possible splicing defect, and one small deletion (Table 2)”; I recommend standardizing the numerical representation (either fully spelled out or alphanumeric) to make the text more harmonious. The same applies to lines 143-145: “Eight (22.86%) were classified as pathogenic variants, 11 (31.43%) as likely pathogenic, 14 (40%) as variants of uncertain significance (VUS), and two (5.71%) as likely benign.” I do not see the tables referenced in the text, nor does the link at the end of the full text for supplementary material work... this is likely an oversight. Nonetheless, I consider the work to be good.

Best regards

Comments on the Quality of English Language

The quality of the English is good, I suggest a revision check for possible typos.

Author Response

Comments and Suggestions for Authors

Dear Authors,

Thank you for the opportunity to review your interesting manuscript. I find the topic compelling and the paper clear and well-structured. I would like to suggest a few minor modifications, as I noticed some references are missing in the text. For example, lines 51-52; 62-64; 71-72; 83-84.

Authors: We appreciate your comments and revision. We revised the references missing in the text and added them in lines 52, 64, 72, and 85.

Line 140: “A total of 35 variants were described: 24 missense, four nonsense, three frameshift, three intronic variants with possible splicing defect, and one small deletion (Table 2)”; I recommend standardizing the numerical representation (either fully spelled out or alphanumeric) to make the text more harmonious. The same applies to lines 143-145: “Eight (22.86%) were classified as pathogenic variants, 11 (31.43%) as likely pathogenic, 14 (40%) as variants of uncertain significance (VUS), and two (5.71%) as likely benign.”

Authors: Thank you for the suggestion. We changed the numerical representations, when suitable in the indicated lines (lines 141 and 144). We also revise and change this error through the manuscript.

I do not see the tables referenced in the text, nor does the link at the end of the full text for supplementary material work... this is likely an oversight. Nonetheless, I consider the work to be good.

Authors: We are very sorry for the error. We had problems adding the tables into the formatted manuscript; therefore, we submitted them along with the supplementary material. 

Best regards

Comments on the Quality of English Language: The quality of the English is good, I suggest a revision check for possible typos.

Authors: We thank the reviewer for the comment, we revised the manuscript and corrected some typos and grammatical mistakes.

Round 2

Reviewer 1 Report

Comments and Suggestions for Authors

The revised version of the manuscript, entitled »Patients with thyroid dyshormonogenesis and DUOX2 variants: molecular and clinical description and genotype-phenotype correlation« submitted to International Journal of Molecular Sciences for a potential publication, has been somewhat improved. The authors have taken most of my suggestions and comments into consideration, albeit briefly. I suggest publication.

Comments on the Quality of English Language

 Minor editing of English language is still required.